Using 3D printed eggs to examine the egg-rejection behaviour of wild birds

Igic Branislav 1 brani.igic@gmail.com
Nunez Valerie 2
Voss Henning U. 3
Croston Rebecca 4 9
Aidala Zachary 2 5
López Analía V. 6
Van Tatenhove Aimee 7
Holford Mandë E. 8
Shawkey Matthew D. 1
Hauber Mark E. 2
1 Department of Biology, University of Akron , Akron, OH , USA
2 Department of Psychology, Hunter College, and The Graduate Center of the City University of New York , NY , USA
3 Citigroup Biomedical Imaging Center, Weill Cornell Medical College , New York, NY , USA
4 Department of Biology, University of Nevada—Reno , Reno, NV , USA
5 Social and Behavioral Sciences Division, Bloomfield College , Bloomfield, NJ , USA
6 Departamento de Ecología, Genética y Evolución, Facultad de Ciencias Exactas y Naturales, Universidad de Buenos Aires , Buenos Aires , Argentina
7 Department of Ecology and Evolutionary Biology, Cornell University , Ithaca, NY , USA
8 Department of Chemistry, Hunter College, The Graduate Center of the City University of New York, and The American Museum of Natural History , NY , USA
9 Department of Biology, The Graduate Center of the City University of New York , NY , USA
Hedrick Ann
Electronic publication date: 2015 May 26
Publication date: 2015
Volume: 3
Electronic Location ID: e965
Received 2015 Mar 28; Accepted 2015 Apr 29
Copyright: © 2015 Igic et al.
Copyright year: 2015
Copyright holder: Igic et al.
License: This is an open access article distributed under the terms of the Creative Commons Attribution License, which permits unrestricted use, distribution, reproduction and adaptation in any medium and for any purpose provided that it is properly attributed. For attribution, the original author(s), title, publication source (PeerJ) and either DOI or URL of the article must be cited.
License URL: https://creativecommons.org/licenses/by/4.0/

Keywords: Artificial egg, Turdus migratorius, Brood parasitism, 3D printing, Egg rejection, American robin, Cowbird, Experimental techniques, Molothrus ater

Funding: NSF #1247550 Air Force Office of Scientific Research FA9550-13-1-0222 Camille and Henry Dreyfus Foundation Funding for this research was provided by the Human Frontier Science Program (RGY83/2012) to MDS and MEH, the offices of the Provost and the Dean of Arts and Sciences at Hunter College to MEH, the Air Force Office of Scientific Research (FA9550-13-1-0222) to MDS, the National Science Foundation to HUV (#0956306) and MH (#1247550), and by the Camille and Henry Dreyfus Foundation to MH. The funders had no role in study design, data collection and analysis, decision to publish, or preparation of the manuscript.

==============================
The coevolutionary relationships between brood parasites and their hosts are often studied by examining the egg rejection behaviour of host species using artificial eggs. However, the traditional methods for producing artificial eggs out of plasticine, plastic, wood, or plaster-of-Paris are laborious, imprecise, and prone to human error. As an alternative, 3D printing may reduce human error, enable more precise manipulation of egg size and shape, and provide a more accurate and replicable protocol for generating artificial stimuli than traditional methods. However, the usefulness of 3D printing technology for egg rejection research remains to be tested. Here, we applied 3D printing technology to the extensively studied egg rejection behaviour of American robins, Turdus migratorius. Eggs of the robin’s brood parasites, brown-headed cowbirds, Molothrus ater, vary greatly in size and shape, but it is unknown whether host egg rejection decisions differ across this gradient of natural variation. We printed artificial eggs that encompass the natural range of shapes and sizes of cowbird eggs, painted them to resemble either robin or cowbird egg colour, and used them to artificially parasitize nests of breeding wild robins. In line with previous studies, we show that robins accept mimetically coloured and reject non-mimetically coloured artificial eggs. Although we found no evidence that subtle differences in parasitic egg size or shape affect robins’ rejection decisions, 3D printing will provide an opportunity for more extensive experimentation on the potential biological or evolutionary significance of size and shape variation of foreign eggs in rejection decisions. We provide a detailed protocol for generating 3D printed eggs using either personal 3D printers or commercial printing services, and highlight additional potential future applications for this technology in the study of egg rejection.

Introduction

Ever since the Nobel-prize winning studies using painted eggshells by Tinbergen (1963), research on avian eggs has focused on studying the biological causes and effects of variation in egg colour, size, shape, and shell strength. Studies of egg rejection examine the behavioural interactions between avian brood parasites and their hosts, with an aim to understand coevolutionary interactions more generally (Dawkins & Krebs, 1979; Rothstein & Robinson, 1998). Brood parasitic birds, such as European common cuckoos Cuculus canorus and North American brown-headed cowbirds Molothrus ater, lay their eggs into nests of other species, prompting these hosts to raise parasitic offspring at the expense of their own offspring (Rothstein & Robinson, 1998; Davies, 2000).

In response to costly brood parasitism, many hosts have evolved the ability to identify and eliminate parasitic eggs from their nests (Davies, 2000; Peer et al., 2005). Extensive experimental work using real eggs of conspecifics and parasites, as well as both real and artificial eggs painted to resemble natural or artificial colours, has revealed the abilities of host species to detect, recognize, and reject parasitic eggs in their nests (Davies & Brooke, 1989; Samas et al., 2014). The cues used by hosts to discriminate own vs. parasitic eggs can include egg size, shape, colour, patterning, or a combination of these traits (Rothstein, 1982; Antonov et al., 2006b; Underwood & Sealy, 2006; Stoddard & Stevens, 2010; Stokke et al., 2010; Zölei et al., 2012). However, parasitic eggs are naturally variable in these traits, and how these subtle differences, particularly in egg size and shape, influence host behaviour has been difficult to test (Lahti & Lahti, 2002; Mason & Rothstein, 1986).

The use of real eggs in egg rejection research is invaluable but also problematic (for discussions see Hauber et al., 2015; Lahti, 2015). The usefulness of real eggs can be limited because the relative influences of egg shape, size, and colour on rejection decisions cannot be easily differentiated (Antonov et al., 2009). The use of natural eggs in experiments can also result in the destruction of viable eggs, which can be problematic from ethical and management considerations. Artificial eggs are used as alternatives to real eggs in experiments and are specifically useful in separating the relative influences of different phenotypic characteristics on rejection decisions (Rothstein, 1982; Hauber et al., 2015; although see Lahti, 2015). However, traditional artificial eggs made of plasticine, plastic, wood, or plaster-of-Paris, are also of limited use (Prather et al., 2007). Their production can be laborious, rely on a single mould or prototype that degenerates with extensive use, and be prone to human error. For example, the mass and quality of plaster cast eggs are strongly influenced by how well a plaster mixture is prepared. As a result, the final products can be constrained in quality, reproducibility, and experimental validity. Indeed, artificial eggs may be rejected based on features that were not experimentally manipulated or controlled (Prather et al., 2007). Finally, traditional methods are not amenable to precise and fine-scale manipulation of egg shape, size, or other characteristics, and thereby reduce the range of hypotheses that can be tested.

3D imaging and printing technologies may help overcome some of the limitations of traditional methods for artificial egg production. 3D printers create plastic objects from digital 3D models, which can be generated using photographs or mathematical formulas (e.g., Troscianko, 2014). 3D printing technology is advantageous over traditional methods for producing artificial eggs because it is less prone to human error and allows precise and controlled manipulation of one, or several, egg traits, including shape, weight, and texture. Therefore, it can both eliminate unwanted variability in mass production of identical stimuli eggs and also allow precise control of how much, and in which parameters, eggs vary. Unlike traditional methods, 3D printing allows the manufacture of hollow eggs, which may imitate characteristics of real eggs more accurately. Lastly, digital 3D models can be easily shared online for other researchers to use, enabling more replicable use of experimental stimuli eggs across different laboratories, continents, and study species, than is possible with traditional methods for making artificial eggs. Despite these potential advantages, the usefulness of 3D printing technology for studying egg rejection decisions remains to be tested.

We applied 3D printing technology in a pilot study to investigate the egg rejection behaviour of American robins Turdus migratorius in relation to brood parasitism by brown-headed cowbirds. Robins lay blue–green immaculate eggs, whereas cowbirds lay smaller beige eggs with spots (Croston & Hauber, 2014). Previous studies using real eggs found that robins generally reject cowbird eggs added to their nests (Briskie, Sealy & Hobson, 1992; Rasmussen, Sealy & Underwood, 2009). Studies using plaster-cast eggs found that robins mostly accept cowbird-sized eggs if they are painted to resemble robin egg colour but reject them if painted to resemble cowbird egg background colour and maculation patterns (Rothstein, 1982; Croston & Hauber, 2014; Kuehn, Peer & Rothstein, 2014; Lang, Bollinger & Peer, 2014). However, cowbird eggs vary greatly in size and shape (18–25 mm length × 15–18 mm breadth; Lowther, 1993), and size can be an important cue for recognition of parasitic eggs by hosts, along with colour (Guigueno, Sealy & Westphal, 2014; Mason & Rothstein, 1986). By placing plaster eggs that varied in size (robin or cowbird egg size), colour (blue–green or white), and maculation pattern into the nests of robins, Rothstein (1982) showed that robins use all three cues in rejection decisions, but are in general more likely to reject eggs if they differ from robin eggs in at least two of these parameters. Although Rothstein (1982) found that robins are more likely to reject white plaster eggs if they are the size of cowbird eggs, rather than the size of robin eggs, it is still unclear whether robins’ rejection decisions vary along the cowbird’s natural gradient of variation for egg shape and size (Mills, 1987; Lowther, 1993). For example, robins may be less likely to reject larger cowbird eggs than smaller cowbird eggs because they are more similar in size to robin eggs. To test this hypothesis, we 3D printed eggs that encompass the natural variation of cowbird egg shapes and sizes to test how these characteristics influence the probability of egg rejection by robins.

Materials & Methods

We performed the following steps to generate 3D printed stimuli for egg rejection research: (1) design 3D digital egg models, (2) 3D print artificial eggs in-house, and (3) upload digital models to a commercial 3D printing service for mass production. We then experimentally tested the utility of 3D printed eggs using American robins as study species.

Design 3D digital egg models

We used the Blender Foundation’s open-source 3D graphics software Blender v.2.70 (freely available from http://www.blender.org/download/) to design 3D model representations of brown-headed cowbird eggs. We designed the initial 3D model using a photograph of a brown-headed cowbird egg (accession no. 13941) sourced from the publicly available digital egg collection of the Museum of Vertebrate Zoology, University of California, Berkeley. To do this, we set the egg image as the background in Blender, first ensuring the egg was vertical (Fig. 1D1), then created a UV Sphere (64 segments and 32 rings), which we expanded to match the width of the cowbird egg (Fig. 1D2). We then used Blender’s Proportional Edit tool to distort the sphere smoothly along its vertical axis with a fall-off proportional to the distance from the centre of distortion. A smaller radius of proportional editing influence resulted in a more “pinched” egg shape whilst a larger radius of influence resulted in a less pointed egg shape. We distorted the sphere until its shape and length matched the background egg (Figs. 1D3–1D4). The resulting ovoid was then resized so that its length matched that of the original egg according to the scale presented in the museum photograph. To create a hollow egg that can be filled with water or gel to better approximate thermodynamic properties of natural eggs (in case this affects heating/cooling rates and thereby egg rejection), a hole was digitally cut at the blunt pole of the egg model such that the printed egg would have a hole no larger than 1–2 mm in diameter. Although 3D printing technology may potentially provide an opportunity to make artificial eggs with shells that can be punctured by hosts with beaks too small to grasp cowbird eggs (Rohwer, Spaw & Røskaft, 1989), testing whether this is possible was not an aim of our study and we encourage future research in this area. The egg model was then exported as a Wavefront object (*.obj) file which could be imported directly into 3D printer software or uploaded to a third party website for commercial printing. Note that since the completion of this field study, we have also succeeded in creating printable egg models in Blender using an XYZ Math Surface where the X, Y and Z values depend on the parameters a, b, and c from the equation for an egg surface modelled by Troscianko (2014), speeding up the 3D model generation process.

Figure 1 Protocol for producing 3D printed replicas of cowbird eggs.

First, a 3D digital egg model was designed from a photograph of a real cowbird egg using Blender software (D1–D4). These digital models were then printed using either a MakerBot Replicator 2X 3D printer (P1–P4) or commercial web-based 3D printing services. The resulting eggs were then painted blue–green or beige. Scale bar: 1 cm.

We also tried an alternate approach for designing digital 3D egg models. We imported the cowbird egg photograph into ImageJ (freely available from imagej.nih.gov/ij) and used the multi-point tool to mark 14 equidistant points on one side of the egg, pole to pole, scaling these points to match the scale bar measurements provided in the egg photograph (23.0 mm in length and 17.7 mm in breadth). We saved these points as “xy coordinates” in a *.txt file, which we then converted into a Microsoft Excel 2010 *.xls document and imported into Autodesk Inventor Professional 2014 (available from http://www.autodesk.com/education/free-software/inventor-professional) as a 2D sketch. The imported points were connected by control vertex-type splines and the resulting half-egg curve closed by insertion of a straight line. To create a solid body from the 2D sketch, the half egg sketch was then revolved one full circle around an axis defined by the previously inserted straight line. The two poles of the egg were respectively too pointed or too blunt, so we manually adjusted the position of the two pole end points to generate the original image’s shape. The resulting sketch was saved as an *.stl file and can be uploaded to websites of commercial 3D printing services (see below).

3D print eggs in-house

We printed the 3D eggs from Acrylonitrile butadiene styrene (ABS) plastic using a MakerBot Replicator 2X 3D printer. The egg object file was sliced for printing with MakerBot MakerWare v. 2.4.1.62 software (Fig. 1P1). We created hollow, water filled, eggs to replicate weights and the thermodynamic properties of similarly sized natural eggs. To replicate the natural variation of egg shape and sizes of brown-headed cowbirds (Lowther, 1993), we scaled the initial egg model to be between 19–24.4 mm in length and 14.4–18.1 mm in breadth using MakerWare (final sizes were measured after painting; see below). Therefore, although all eggs were designed from the same initial cowbird egg, each resulting printed egg was unique in shape and size to replicate the diversity of cowbird egg phenotypes and avoid potential biases of pseudo replication (Hurlbert, 1984). We printed eggs with four internal shell layers built on top of each other, using 0.3 and 0.2 mm layer thicknesses, and 0% infill (Fig. 1P2). We filled these eggs with water using a syringe and then sealed them using thick ABS-acetone glue made by dissolving ABS in acetone to create a thick paste. We then diluted this ABS-acetone glue using more acetone and coated the entire egg surface to ensure that the product was watertight. Egg surfaces were sanded smooth prior to painting and between paint layers when necessary. Whole eggs can also be submerged in 100% acetone (Fig. 1P3) to create a smooth surface (Fig. 1P4).

We painted eggs blue–green and beige using two coats of nontoxic acrylic or latex house paint (Behr PREMIUM PLUS™ Interior Paint; Behr, Santa Ana, California, USA), following Croston & Hauber (2014). We measured the reflectance of painted artificial eggs, and the background coloration of real robin and cowbird eggs, following methods described in Igic et al. (2012). The background coloration of natural cowbird eggs is difficult to replicate using commercial paint designed for human vision (Croston & Hauber, 2014). Therefore, we used paint that provokes robins to reject artificial eggs at similar rates as they do natural cowbird eggs, and which is similar to the beige background coloration of cowbird eggs between 400–700 nm (Fig. 2B; Briskie, Sealy & Hobson, 1992; Croston & Hauber, 2014). American robins do not appear to prioritise ultra-violet wavelengths over other avian-perceived wavelengths in egg rejection decisions, but rather use the whole avian visible spectrum to identify foreign eggs in their nests (Croston & Hauber, 2014). The effect of maculation was excluded from our study because it can be used as an additional, or separate, cue to eggshell background coloration for identifying parasitic eggs by robins and other species (Rothstein, 1982; Stoddard & Stevens, 2010).

Figure 2 Summary of artificial egg colours used in experiments and their rate of rejection by robins.

(A) Photographs of natural American robin nests containing an introduced cowbird-sized blue–green (top) or beige (bottom) painted 3D printed egg. (B) Average reflectance of natural robin eggs, natural cowbird eggs, and 3D printed eggs painted blue–green or beige to respectively resemble robin or cowbird egg colours. (C) Proportion of blue–green or beige painted eggs (of all sizes) that were rejected by robins. Numbers above bars illustrate sample sizes.

We measured the weight of the water-filled eggs immediately after sealing/painting and again 12 days later to ensure that the water did not evaporate or leak; the average weight loss was 1% (range: 0.1–2.4%; n = 38) with two additional eggs losing 7% and 15%, respectively: these eggs were discarded from the experiment. In addition to creating hollow water filled eggs, we were able to print solid filled eggs using two internal shell layers with 0.3 mm layer thickness and 100% infill.

In addition to printing variably sized eggs, we quantified the precision of 3D printing eggs by printing 80 eggs of a set size (22.6 mm length × 18.0 mm breadth) through Shapeways (see below) and measuring their size dimensions to the nearest 0.1 mm using digital callipers (Mitutoyo Digimatic Plastic Callipers model 700–126; Mitutoyo, Kanagawa, Japan) and weight to the nearest 0.1 g using an American Weigh AMW-155 (150 g × 0.1 g) scale. We compared these measurements to those taken on plaster eggs produced following Croston & Hauber (2014) using coefficients of variation.

Upload digital models to commercial 3D printing service

To provide researchers with access to 3D printed cowbird eggs without a requirement for personal 3D printers or 3D modelling software, we provide our egg model designs (File S1), allowing other researchers to use our 3D designs with commercial 3D printing services. To test the utility of using commercial 3D printing services, we uploaded our designs to www.shapeways.com. We chose “white strong and flexible plastic, polished” as the manufacture material, because the eggs were printed solid filled and their resulting weight (3.1 g) was similar to the average weight of brown-headed cowbird eggs (mean: 3.2 g, range: 2.5–3.8; Ankney & Johnson, 1985). For ease of access, our egg design can also be directly printed through Shapeways (https://www.shapeways.com/shops/VN).

Experiment using 3D printed eggs

To test the utility of 3D printed eggs, we studied the egg rejection behaviour of American robins during May–July 2014, in and around Ithaca, Tompkins County, NY, USA. Robins are highly commensal with humans (Vanderhoff, Sallabanks & James, 2014), and often nest on human-made structures or in vegetation nearby. We located nests by searching in and around residential areas, parking lots, and natural woodland areas (Croston & Hauber, 2014). Nests were deemed active if natural clutch size increased between subsequent visits or if a robin was observed to be incubating the eggs. Upon finding an active nest that contained two or more robin eggs, we placed a 3D printed egg that was painted blue–green to resemble colour of robin eggs (n = 14) or beige to resemble the background colour of cowbird eggs (n = 14), allocated at random. The timing of parasitism, with respect to whether the female is laying or incubating eggs, does not affect probability of egg rejection in this species (Croston & Hauber, 2014), so we placed eggs in nests at either of these stages. We did not remove any host eggs during the experiment because removing a single egg has no effect on the rejection responses of robins (Briskie, Sealy & Hobson, 1992). Every nest received an artificial egg with a unique size or shape.

We monitored all nests daily for six days following parasitism, and visually determined the presence or absence of artificial eggs. Robins generally reject eggs within five days following parasitism (Croston & Hauber, 2014) by grasping them in their bills and carrying them away from the nest (Video S1; Rothstein, 1975; Rasmussen, Sealy & Underwood, 2009). By using a large grasp-ejector species in our experiments, such as the American robin, we avoided potential spurious results of egg acceptance because artificial eggs cannot be pierced (Martín-Vivaldi, Soler & Møller, 2002; Boulton & Cassey, 2006; Soler et al., 2015) and isolated rejection probabilities to the recognition of parasitic eggs by hosts (Hauber & Low, 2014; Mendelson, 2015), rather than physical constraints of rejecting eggs of particular size or shape. Indeed, our specific aim was to examine how egg size and shape influence robins’ egg rejection decisions, rather than the physical constraints on egg rejection. Although an ideal study species would be a grasp-ejector that is known to respond to the size and shape of cowbird eggs, we are unaware whether such a host has been identified. We considered rejection to have occurred when the artificial egg was absent from the nest, but the nest showed no signs of partial or total predation, such as missing or broken robin eggs. If the female had not been seen incubating for the past three days and the eggs were cold, we considered the artificial egg to have been rejected via nest abandonment. A single nest was abandoned following experimental parasitism with a beige egg, whereas no nests were abandoned following experimental parasitism with a blue–green egg. We considered acceptance to have occurred if the artificial egg was still present in the nest after six days following experimental parasitism. We excluded any experimental nests from our analysis where all the robin’s eggs hatched prior to this six day acceptance period and the artificial egg was still present in the nest. For nests that reached an experimental outcome (either rejection or acceptance of artificial eggs) before eggs began to hatch, we parasitized those nests again with an artificial egg of the alternate colour immediately after determining acceptance or rejection of the previous egg. This was done to maximise the power of our analyses through a repeated-measures experimental design. In this way, we successfully completed experiments at six nests with two artificial eggs in succession and 16 nests with a single artificial egg. In addition, five nests were predated following parasitism and five nests had eggs that hatched before an experimental outcome was reached, and were excluded from the analyses.

For each experiment we recorded the colour, length, and breadth of artificial eggs, whether the female was flushed off the nest during parasitism, and the order in which artificial eggs were presented. Prior work showed no effect of presentation order on the outcome of egg colour experiments in this population (Croston & Hauber, 2014); nevertheless, we still tested for an order effect statistically (see below).

Handling of wild birds, including nests and eggs, were carried out in accordance with Federal and State guidelines and laws. Work conducted on private properties was with consent of the affected landowners, and approved by the Institutional Animal Care and Use Committee of Hunter College, City University of New York (No. MH 2/16-T3) and US bird banding laboratory (no. 23681).

Statistical analysis

We used penalized-likelihood models to test whether rejection of artificial eggs was affected by their colour, size, shape, order of presentation, or whether the female was flushed off the nest when eggs were added. To examine whether artificial egg shape and size affected rejection responses, we first used principal component analysis on egg length and breadth measurements to produce two orthogonal axes: PC1 explained 99% of the total variance and compared average egg length (loading: −0.83) and breadth (loading: −0.55), and therefore was a measure of total egg size (Fig. 3C); PC2 explained 1% of the total variance and compared egg length (loading: 0.55) to egg breadth (loading: −0.83), and therefore was a measure of egg shape (Fig. 3C). There was a complete separation in the response outcomes of our data in regards to egg colour (all blue–green eggs were accepted); therefore, we used Firth’s penalized-likelihood logistic regression for all tests and tested each factor separately while also controlling for nest ID. Not only is penalized-likelihood logistic regression useful in overcoming problems of complete separation, but it outperforms traditional logistic regression when sample sizes are small to medium (Heinze & Schemper, 2002). We used rejection response (binary outcome: accept/reject) as an independent variable and fit each factor (colour, egg size, egg shape, etc.) as explanatory variables. Removing a nest that was abandoned following experimental parasitism with a beige egg did not affect our results (Soler et al., 2015). We tested whether flushing females off of their nest during simulated parasitism affected their probability of rejecting beige eggs using a Fisher’s exact test (Hanley et al., 2015). We present mean odds ratios and 95% confidence intervals for all tests.

Figure 3 Factors that did not influence rejection of artificial eggs by robins in our experiments

(A) Acceptance and rejection outcomes for 3D printed eggs in relation to their size and shape. Brown-headed cowbird eggs vary between 18–25 mm in length and 15–18 mm in breadth (Lowther, 1993). Grey dotted lines represent the directions of principal component axes. Proportion of eggs rejected by robins in relation to (B) the order of presentation; (C) whether the female was flushed off the nest during parasitism; (D) both colour and make of artificial eggs (plaster vs. 3D printed), with plaster egg data sourced from Croston & Hauber (2014). Numbers above bars illustrate sample sizes.

In addition to testing the effects of egg colour, we tested whether rejection of artificial eggs by robins is influenced by whether eggs are made using 3D printing or plaster-of-Paris. Here we compared our rejection data of 3D printed eggs to rejection data of a recent study that investigated egg rejection behaviour using the same robin population and paint colours, but using 21.4 mm length × 16.4 mm breadth eggs made of plaster (Croston & Hauber, 2014). This included robins’ responses to 13 blue–green plaster eggs and 19 beige plaster eggs. We fit a penalized-likelihood logistic regression with egg rejection response as the independent variable; nest ID, egg colour, egg make, and the interaction between egg colour and make as fixed effects. The interaction was not significant χ12=0.46,P=0.5, so we refit the model without this interaction. All penalized-likelihood logistic regressions were fit using the logistf package (Ploner et al., 2013), and all statistical tests were two-tailed (α = 0.05) and conducted using R v 3.0.1 (R Development Core Team, 2013).

Results

American robins accepted all eggs painted to resemble robin egg colour and rejected 79% of eggs painted to resemble cowbird egg colour (Video S1), but we detected no effects of egg size or shape on rejection probability (Figs. 2 and 3). Robins were more likely to reject beige eggs than blue–green eggs (beige vs. blue–green: odds ratio = 15.0, 95% CI [1.3, 425.7]; χ12=4.79, P = 0.029; Fig. 2C). Egg size (PC1: odds ratio = 1.5, 95% CI [0.8, 3.1]; χ12=1.5, P = 0.22; Fig. 3A), egg shape (PC2: odds ratio = 1159.5, 95% CI [0.0001, >10,000]; χ12=0.72, P = 0.40; Fig. 3A), the order in which eggs were presented (1st vs. 2nd: odds ratio = 3.1, 95% CI [0.4, 36.4]; χ12=1.10, P = 0.30; Fig. 3B), and whether the female was flushed off the nest when the artificial egg was added (no vs. yes: odds ratio = 0.1, 95% CI [0.001, 2.2]; χ12=1.92, P = 0.17; Fig. 3C) did not influence rejection probability in our study. When considering beige eggs alone, size (PC1: odds ratio = 1.2; 95% CI [0.6, 3.1]; χ12=0.15, P = 0.7; Fig. 3A) and shape (PC2: odds ratio = 6.1; 95% CI [0.04, 848.1]; χ12=0.56, P = 0.45; Fig. 3A), and whether the female was flushed (odds ratio = 0; 95% CI [0, 10.4]; Fisher’s exact test P > 0.99) did not influence their probability of being rejected.

Next, we compared our data on the rejection of variably sized 3D printed eggs to data from a recent study that examined egg rejection behaviour of the same robin population using the same paint colours, but invariant artificial eggs made of plaster (Croston & Hauber, 2014). Egg colour (beige vs. blue–green: odds ratio = 14.2; 95% CI [1.6, 291.5]; χ12=5.9, P = 0.02; Fig. 3D), but not manufacture type (plaster cast vs. 3D printed: odds ratio = 1; 95% CI [0.4, 2.5]; χ12<0.01, P > 0.99; Fig. 3D), was a significant predictor of rejection response.

The size dimensions and weight of eggs printed the same size were low in variance, and lower than that of same sized plaster-cast eggs (Table 1). In particular, 3D printing was much better at producing artificial eggs with a consistent weight (Table 1), perhaps because the weight of plaster eggs is strongly dependent on how the plaster mixture is prepared. 3D printing was also more precise in reproducing a set size and shape of eggs than the plaster method (Table 1).

Table 1 Coefficients of variation expressed as percentages for size and weight of 3D printed and plaster-cast artificial eggs.

	n	Length	Breadth	Aspect ratio	Weight	
3D printed	80	0.54%	0.73%	0.82%	0.90%	
Plaster-cast	63	2.34%	0.28%	2.39%	15.00%	

Discussion

Here, we demonstrate the utility of 3D printing technology in egg rejection studies. We provide a detailed protocol to enable other researchers to use the economical and flexible 3D design and printing technologies in their own research. Our experimental findings correspond with previous studies showing that colour is as an important cue for rejection of cowbird eggs by robins (Rothstein, 1982; Croston & Hauber, 2014; although see Lorenzana, Sealy & Murphy, 2002). Although previous studies showed that robins are more likely to reject eggs smaller than their own (Rothstein, 1982), our results suggest that egg rejection decisions of robins do not vary across the natural variation of cowbird egg shapes and sizes. However, our sample sizes were limited and a more extensive experiment is needed to detect weak effects. The lack of statistical effects of parasitic egg shape and size are consistent with robins rejecting spherical and elongate egg shapes from their nests at similar rates (Underwood & Sealy, 2006) and cowbirds parasitizing host species’ nests indiscriminately with respect to host egg size (Mills, 1987). The precise and controlled manipulation of stimulus characteristics using 3D printing technology showcased here will enable others to expand upon our findings and test similar recognition mechanisms in other host species.

3D printing permits precise, reproducible, and accurate manipulation of stimuli characteristics at a level that cannot be achieved by traditional methods for making artificial eggs. However, 3D printed eggs may not necessarily produce more biologically meaningful results than traditional artificial eggs, but rather expand the range of hypotheses that can be tested. 3D printing enables researchers to apply a more standardized and replicable technique in production of artificial eggs as evidenced by lower variability when producing identical stimuli, and enables much finer control over how much, and in which characteristics, stimuli vary. Furthermore, in contrast to traditional methods, 3D printing allows the replicable use of experimental stimuli across studies because digital 3D egg models can be exchanged between researchers. For example, we provide our design schematics of cowbird eggs (File S1) that can be used by others in combination with personal or commercial 3D printers. Commercial 3D printing services also allow manipulation of characteristics of uploaded 3D image files, and thereby allow researchers to design egg stimuli for their specific needs without the need of a personal 3D printer. We conducted proof of concept experiments using these commercially produced 3D printed cowbird eggs and found that they can also be accepted when painted to resemble robin egg colour (n = 2) and rejected when painted to resemble cowbird egg background colour (n = 1). Although 3D printing does not necessarily overcome the potential limitations of using artificial stimuli in behavioural research (Lahti, 2015), it is an important step towards creating more natural-like artificial stimuli to reduce potential biases.

Beyond the uses demonstrated here, 3D printing has the potential for widespread applicability across many areas of behavioural ecology, comparative psychology, and neuroethology. For brood parasitism research, 3D printing may enable more precise examination of the sensory and motor controls of, and their constraints on, egg recognition and rejection (Rothstein, 1982; Antonov et al., 2009). It allows precise manipulation of egg shape and potentially, shell thickness, making it potentially possible to create egg shapes that can, or cannot, be grasped and removed from the nest (Rohwer & Spaw, 1988) or eggs with shells that are thin enough, or too thick, for hosts to reject by puncturing the shell (Antonov et al., 2006a; Spottiswoode, 2010). Indeed, when artificial eggs cannot be pierced by small hosts, rejection of eggs by piercing their shells may not be possible, and thereby generate spurious results of egg acceptance or nest abandonment, instead of egg ejection (Martín-Vivaldi, Soler & Møller, 2002; Boulton & Cassey, 2006; Soler et al., 2015). The capacity for 3D printing to produce artificial eggs that can be rejected by hosts via puncturing their shells may be possible in the future, with the advent of light and brittle printable materials; however, this concept remains to be tested. Weight to size ratio and surface texture can also be altered to test how these properties affect egg rejection behaviour. More broadly, 3D printing is likely to revolutionize behavioural research by allowing precise manipulation and economical production of controlled stimuli. Potential future applications include, but are not limited to, studies of visual recognition of predators, prey, and conspecifics (Nelson, 2010); sexually selected visual signals (Callander et al., 2013); and the evolution of egg morphology (Preston, 1953).

Supplemental Information

File S1 Digital 3D model representation of a brown-headed cowbird egg

Click here for additional data file.

Video S1 Responses of a female American robin to 3D printed eggs in her nest

Responses of a female American robin following the introduction of a 3D printed egg (either painted blue–green to resemble robin egg color or beige to resemble cowbird egg color) to her nest.

Click here for additional data file.

Data S1 Dataset

Dataset 1. Data collected during rejection experiment and for size dimensions of 3D printed and plaster-cast eggs.

Click here for additional data file.

We thank Cornell University, Ithaca College, and the many landowners that allowed us to conduct our research on their property. We thank the Shawkey lab for helpful comments.

Additional Information and Declarations

Competing Interests

Author Contributions

Animal Ethics

Field Study Permissions

The authors declare there are no competing interests.

Branislav Igic performed the experiments, analyzed the data, wrote the paper, prepared figures and/or tables, reviewed drafts of the paper.

Valerie Nunez, Henning U. Voss, Rebecca Croston, Zachary Aidala, Mandë E. Holford, Matthew D. Shawkey and Mark E. Hauber conceived and designed the experiments, wrote the paper, reviewed drafts of the paper.

Analía V. López and Aimee Van Tatenhove performed the experiments, wrote the paper, reviewed drafts of the paper.

The following information was supplied relating to ethical approvals (i.e., approving body and any reference numbers):

ACUC: Hunter Institutional Animal Care and Use Committee no. MH 2/16-T3.

The following information was supplied relating to field study approvals (i.e., approving body and any reference numbers):

US Bird Banding Laboratory no. 23681.

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
