# Peer review of "Using 3D printed eggs to examine the egg-rejection behaviour of wild birds"

_PeerJ, doi:10.7717/peerj.965_

## Round 0.1 · original submission · Minor Revisions

Please pay especial attention to the comments of Reviewer 2. It would be best if you could expand the paper to include more detail on the generation of hollow eggs, and to include information on whether hollow eggs can be printed from plastics that make them possible to puncture with bill strikes that resemble those of known puncture-ejectors.

Reviewer 1 ·

Basic reporting

No Comments (= criteria fulfilled)

Experimental design

No Comments (= criteria fulfilled)

Validity of the findings

No Comments (= criteria fulfilled)

Additional comments

The main message of this work is important and provides a new methodological avenue for the future studies of host responses to brood parasite eggs. I fully agree with authors about the limitations of traditionally hand-made egg models (Introduction) and I applaud them for pioneering the use of 3D printing technology in the study of brood parasite-host interactions (and perhaps beyond, l. 336-338).

Specific comments:

36: remove “Grim 2007” (that is not a work about eggs), use instead Samas et al. 2014 (Front. Zool. – the use of both natural conspecific and artificial cuckoo-like eggs; also useful reference for assessing nest abandonments with control nests, see below), for example.

41: “difficult to test” – actually, did anybody test effects of individual variation in parasite eggs? Perhaps Lahti and Lahti 2002 (Anim. Behav.) is a suitable reference here?

45: “confounded” with what? You mean “inter-correlated”?

79: “resemble cowbird egg background colour” – discuss spotting too, please.

114: A very good point!

146: Excellent.

159: Can you be more specific? Is this because robins use UV-component of egg reflectance to accept/reject the foreign egg?

209-212: Normally one would need a control group of nests to test whether nest abandonment represents a specific response to parasitism; but as only 1 experimental nest was deserted, the control group is not necessary.

217: “opposite” is not a correct word, I guess.

217: Explain reasons for repeated tests.

225: This is interesting because a recent study (which is not cited) showed that flushing may indeed affect host responses: Hanley et al. 2015 (Sci. Rep.). That study should be referred to as a rationale for including “flushing” as a predictor in your models (otherwise it is unclear why this predictor was included).

310: Yes, this is a fundamental point: this may ameliorate one of the serious problems of brood parasitism studies: each study group seem to use idiosyncratic model eggs which prevents exact replication of methods and introduces noise into meta-analyses.

315-317: I am lost here – explain what you mean here.

327: Excellent idea.

463-465: This only repeats info already given in fig. 3a itself.

·

Basic reporting

See below.

Experimental design

See below.

Validity of the findings

See below.

Additional comments

Using 3D printed eggs to examine the egg-rejection behavior of wild birds

Review:

I find this a difficult paper to review for PeerJ because it is a composite of valuable new techniques for making artificial eggs and a report of results on the importance of variation in size and color of artificial eggs using American Robins. However, the latter seems relatively unimportant given past experiments on robin egg rejection and the small sample of eggs that were rejected because they were painted beige. Using 3D printing to generate artificial eggs and providing the attendant formulae, equations and instructions for doing are indisputably valuable. The results of the field experiments however, verge on uninformative replication of old experiments.

Without doubt these new results better control for variation in size among real cowbird eggs, but assessing that variation in such a large grasp ejector seems uninformative. The results mirror those obtained long ago by Rothstein, Sealy, and others: size and shape variation in artificial cowbird eggs did not affect rejection, but background color did. Had the rejecter species used for these tests been a host that is only marginally able to reject cowbird eggs, then assessing the importance of variation in egg size and shape on rejections would be informative, but robins can so easily handle all of the size variation found in cowbird eggs that this additional test seems scarcely worth reporting. In this sense, the experimental work seems to verge on “repetition of well known and widely accepted results,” which is against PeerJ policy.

Certainly all of the material on 3D printing seems exceptionally valuable, and it would be lovely if this could be expanded to assess whether the plastic used to generate these eggs can be punctured. Thus, my preference would be to see this paper revised just to present the instructions for using 3D printers for generating experimental eggs. There have been so many egg addition experiments, using all sorts of eggs, that there seems to be no reason to critically test whether or not artificial eggs generated using 3D printing can be used to successfully elicit ejection behavior, and using robins to assess the importance on natural variation in egg size of cowbirds, while convenient for generating good samples, offers to little that is new to seem worthwhile. Indeed, the title suggests that it is the new data on ejection behavior that will be the focus of the paper, but the real value of the paper will be the demonstration that 3D printing can be used to make experimental eggs and the descriptions and tables needed for doing this.

My immediate impression was that this manuscript would have solved (brilliantly!) the problem of generating artificial eggs that could be used to test ejection in species with bills and body sizes too small to grasp-eject cowbird eggs. For work with these species we desperately need artificial eggs that can be puncture ejected; further, we need artificial eggs for which shell thickness and roundness, both of which affect puncture resistance, can be varied so that the importance of these variables can be assessed in experiments with puncture ejecting species. Such eggs would be valuable for hosts of cowbirds, and exceptionally valuable for work with hosts of the common cuckoo. In cuckoo hosts of intermediate size, selection for puncture ejection is intense because no host chicks survive if the cuckoo hatches in a timely fashion, but host species vary greatly in bill shapes, which strongly affect their success at puncture attempts, especially for hosts that are too small for grasp ejection.

Thus, I was distressed to see most of the paper devoted to the generation of artificial eggs that were only appropriate to work with grasp ejectors, or that were only evaluated with a grasp ejecting species. At the same time, the authors tantalizingly report that their 3D printed eggs were hollow and could be filled with water or gels that would mimic real cowbird eggs. Expanding the paper to include far more detail on the generation of hollow eggs, and to include information on whether hollow eggs can be printed from plastics that make them possible to puncture with bill strikes that resemble those of known puncture-ejectors would enormously increase its value. Here, proof that such eggs actually work, possibly by using house wrens as model puncture ejectors, would be a great addition, and not seem unnecessarily redundant, as do the results presented here on egg size variation using a large grasp ejector.

Sievert Rohwer

Reviewer 3 ·

Basic reporting

No comments

Experimental design

No comments

Validity of the findings

No comments

Additional comments

This paper deals with a subject that would be of interest to readers of Peer J and also deals with a subject that excites controversy within the egg-recognition of birds.

This is a nice paper. I think that is an interesting and useful contribution to the literature about the behavioural interactions between avian brood parasites and their hosts. In general, the writing is clear. The arguments are clearly presented and the interpretation of the results justified.

---

## Round 0.2 · accepted · Accept

Thank you for your conscientious changes.